# Predicting biomass of resident kōkopu (*Galaxias*) populations using local habitat characteristics

Ben R. J. Crichton[1,2]*, Michael J. H. Hickford[1,3], Angus R. McIntosh[2], David R. Schiel[3]

**1** Marine Ecology Research Group, School of Biological Sciences, University of Canterbury, Christchurch, New Zealand, **2** Freshwater Ecology Research Group, School of Biological Sciences, University of Canterbury, Christchurch, New Zealand, **3** National Institute of Water and Atmospheric Research, Christchurch, New Zealand

☯ These authors contributed equally to this work.
* bcrichtonnz@gmail.com

**Data Availability Statement:** We have created and linked a Figshare folder that includes the dataset. link: https://figshare.com/s/

## Abstract

With the global decline of freshwater fishes, quantifying the body size-specific habitat use of vulnerable species is crucial for accurately evaluating population health, identifying the effects of anthropogenic stressors, and directing effective habitat restoration. Populations of New Zealand's endemic kōkopu species (*Galaxias fasciatus*, *G. argenteus*, and *G. postvectis*) have declined substantially over the last century in response to anthropogenic stressors, including habitat loss, migratory barriers, and invasive species. Despite well-understood habitat associations, key within-habitat features underpinning the reach-scale biomass of small and large kōkopu remain unclear. Here, we investigated whether the total biomass of large (> 90 mm) size classes of each kōkopu species and the composite biomass of all small ($\leq$ 90 mm) kōkopu were associated with components of the physical environment that provided refuge and prey resources across fifty-seven 50-m stream reaches. Because kōkopu are nocturnal, populations were sampled by removal at night using headlamps and hand-nets until reaches were visually depleted. Based on Akaike's information criterion, greater large banded kōkopu biomass was most parsimoniously explained by greater pool volume and forest cover, greater large giant kōkopu biomass by greater bank cover and pool volume, and greater large shortjaw kōkopu biomass by greater substrate size and pool volume. In contrast, greater composite small kōkopu biomass was best explained by smaller substrate size, reduced bank cover, and greater pool volume. Local habitat associations therefore varied among kōkopu species and size classes. Our study demonstrates the importance of considering the ontogenetic shift in species' habitat use and provides an effective modelling approach for quantifying size-specific local habitat use of stream-dwelling fish.

686b54046be999b5ed95 DOI: 10.6084/m9.
figshare.17203961.

**Funding:** BRJC, MJHH and DRS were financially
supported by the NZ Ministry of Business,
Innovation and Employment (C01X1615; https://
app.dimensions.ai/details/grant/grant.7565396).
Publication costs were offset by the University of
Canterbury's Open Access Fund. The funders had
no role in study design, data collection and
analysis, decision to publish, or preparation of the
manuscript.

**Competing interests:** The authors have declared
that no competing interests exist.

## Introduction

Given the widespread decline of freshwater fishes [1], it is crucial to quantify which habitats are used during all stages of a species' life cycle so that population health can be evaluated accurately, effects of anthropogenic stressors can be tested, and successful rehabilitation measures implemented [2]. Anthropogenic stressors such as pollution, habitat fragmentation and degradation, introduced species, river regulation, and over-exploitation have contributed to a substantial decline in riverine fish populations over the last century [3]. Unfortunately, statistical models used for delineating the effects of anthropogenic stressors on fish populations may be inaccurate or limited if they are calibrated using only a fraction of the habitats used by a species during its lifecycle [4]. Without accurate models relating body size and specific habitats, population assessments may be biased, which could lead to ineffective management decisions and unsuccessful, wasteful, or even harmful restoration efforts by excluding important microhabitats such as spawning sites or nursery grounds [5, 6].

Influential habitat variables that often determine the habitat selection of stream-dwelling fish include water velocity, in-stream refuges, and overhanging vegetation [7]. Often, pools are preferentially used microhabitats for drift-feeding fish because they have slower water velocities that typically reduce an individual's energetic expenditure while improving feeding efficiency [8]. In-stream cover, such as undercut banks, root-wads, debris dams, and interstices between large substratum components are important refuges on which many fish rely to minimise the risk of predation and the impacts of physical disturbances [9]. Additionally, overhanging vegetation, such as riparian vegetation or forest canopy cover, is linked to a stream's primary productivity and plays a crucial role in providing terrestrial subsidies, in-stream cover, and hydrological stability [10, 11]. Therefore, these habitat features are likely influential determinants of habitat selection during at least one stage of the lifecycle of stream-dwelling fishes.

The importance of specific habitat features on habitat selection is often also strongly determined by body size [12]. In freshwater fishes, variation in body size-related habitat selection is typically due to individual selection of microhabitats that maximise energy gain and minimise energy expenditure or increase survival [13, 14]. Although body size is closely related to habitat use, it is important to account for the life-history characteristics of the species examined, because species that are morphologically and phylogenetically similar may still respond differently to microhabitat characteristics [15, 16]. For species in which different size classes inhabit the same local environment (e.g., the same stream reach), restoration efforts should incorporate potential ontogenetic shifts in size-specific microhabitat requirements to account for all life-history stages in an ecosystem. This is especially important for species that exhibit intraspecific or intra-family competitive hierarchies, because inferior individuals may avoid preferred habitats when dominant congeners are present [17]. Social competitive hierarchies in freshwater fish often follow a size-related structure, with large dominant individuals monopolising key feeding habitats and smaller individuals being displaced to less advantageous habitats [18, 19]. Therefore, understanding how abiotic and biotic influences affect the habitat use of distinct size classes is essential to obtain a robust evaluation of population habitat use.

New Zealand's endemic banded kōkopu (*Galaxias fasciatus*), giant kōkopu (*G. argenteus*), and shortjaw kōkopu (*G. postvectis*), hereafter collectively referred to as 'kōkopu', are diadromous fishes that inhabit freshwater environments during all but their larval life stage. Over the last century, kōkopu have undergone considerable declines in response to the loss and degradation of adult habitats through activities including drainage schemes, land-use changes, migratory barriers, and deforestation [20–22]. Additionally, introduced species like trout alter

kōkopu habitat selection through predation and competitive exclusion [23]. Post-larval kōkopu are also harvested in the culturally, recreationally, and commercially important whitebait fishery [24]. Despite population declines, it is unknown how these anthropogenic stressors specifically alter kōkopu populations, due to the lack of accurate size-specific habitat models.

Although size-specific habitat models have not been developed for kōkopu, there is a good understanding of their general distribution and habitat use [25]. At landscape scales, kōkopu typically occupy relatively unmodified catchments accessible from the sea, so these factors will be important to account for before local habitat influences can be considered. Greater local kōkopu densities are often associated with the availability of slow-flowing pools because kōkopu are opportunistic, mostly nocturnal predators, that rely on the transport of aquatic and terrestrial invertebrates into pools from fast-flowing upstream habitats [20, 26]. Banded and shortjaw kōkopu are forest specialists, rarely inhabiting streams without forest canopies, but giant kōkopu also inhabit estuaries, swamps, or ponds [25, 27]. Each species depends on refuge areas for secure diurnal resting, predator escapement, and shelter from flood events [28]. Despite having slightly different niches, kōkopu species commonly co-occur and share similar microhabitats and diets. This indicates that each species should be influenced similarly by local habitat characteristics.

Although juvenile and adult kōkopu generally occupy the same stream reaches, individual microhabitat selection is strongly determined by the presence of larger conspecifics or congenerics [29]. For example, small giant kōkopu minimise agonistic interactions with larger dominant conspecifics, which control large pools at night, by feeding during the day or by occupying alternative microhabitats at night [30]. Similarly, large banded kōkopu use deep, slow-flowing pools with coarse substratum, whereas smaller individuals are likely displaced into shallow pools with faster water velocities and finer substratum [31]. Although size-related kōkopu microhabitat segregation [19, 32, 33] and the influence of habitat characteristics on total kōkopu biomass [7, 17] are understood, the influence of local habitat characteristics on the reach-scale biomass of small and large kōkopu separately is unknown. Such information would provide a more comprehensive and accurate description of kōkopu habitat use that could be used to improve habitat restoration efforts. Additionally, by understanding how small and large kōkopu are influenced by local environments, while all other influential environmental variables are being controlled for, a standardised prediction of likely kōkopu biomass based solely on local habitat characteristics can be obtained. These standardised estimates will allow the isolation and accurate testing of how individual environmental manipulations including dispersal barriers, introduced predators, fishing pressure, or habitat restoration efforts affect kōkopu populations by controlling for habitat-related biases.

We aimed to identify habitat characteristics that influence the biomass of small and large banded, giant, and shortjaw kōkopu species endemic to New Zealand streams. Specifically, we evaluated how a candidate set of models that included combinations of local habitat features performed in explaining variation in the biomass of each species' large size class and the composite biomass of all small kōkopu. To achieve this objective, kōkopu populations were surveyed across physically diverse stream reaches. We predicted that for each kōkopu species: (1) large kōkopu biomass would increase with pool volume, whereas small kōkopu biomass would decrease in such habitats, putatively due to larger fish competitively excluding smaller individuals within these key feeding areas; (2) large and small kōkopu biomass would increase with increasing bank cover and substrate size due to both providing refuges to all size classes; and (3) both large and small kōkopu biomass would increase with forest canopy cover due to it likely providing greater food availability.

## Methods

### Study sites

To investigate which habitat features are most strongly associated with reach-scale kōkopu biomass, three 50-m reaches were sampled within each of 19 streams on the West Coast of New Zealand's South Island during May and June 2021. Although kōkopu species typically spawn between May and July, it is unlikely that this alters the representativeness of our findings because kōkopu generally spawn on the stream margins within their home ranges [34, 35] and because there were no notably fecund or spent individuals observed. However, there is some uncertainty around the spawning behaviours of kōkopu, particularly giant kōkopu, with some fish appearing to migrate downstream to spawn [36]. Local topographic maps, site visits, and databases, such as Freshwater Ecosystems of New Zealand (FENZ; [37]) and the New Zealand Freshwater Fish Database (NZFFD; [38]), were used to select streams that were within the range of each kōkopu species and that had no fish passage barriers or introduced species. Selected streams were open to whitebait fishing, located in minimally degraded catchments, within 17 km of the coast, and < 100 m in elevation to control for landscape-scale influences. Physically diverse stream reaches were selected to provide a robust understanding of how individual habitat variables influenced kōkopu biomass. Sampling took place within two months to minimise seasonal differences in kōkopu biomass.

### Habitat survey

Study reaches, beginning and ending at riffles which acted as minor fish barriers between reaches, were in areas with minimal surface turbulence or natural visual-obstruction deposits (e.g., foam or fine debris collections), and were no deeper than 1.5 m. Habitat surveys, completed during daylight hours, involved measuring pool volume, in-stream bank cover, average substrate size, and percentage cover of forest canopy within each reach. Forest cover was measured at approximately eight locations within each reach using a convex spherical crown densiometer [39] while standing in the middle of the waterway and facing upstream. In-stream bank cover was recorded by measuring the perimeter of root wads, undercut banks, or debris dams accessible to fish. Pool volume was calculated using:

$$PV = \left(\frac{W}{2}\right) \times \left(\frac{L}{2}\right) \times \pi \times D \qquad \text{(Eq 1)}$$

where $PV$ is pool volume (m³), $W$ is maximum pool width (m), $L$ is maximum pool length (m), and $D$ is average pool depth (m). Note that $D$ was calculated from ten depth measurements along the $W$ axis. Average substrate size within each reach was calculated from approximately 60 stones randomly selected using a Wolman's walk [40].

### Kōkopu biomass survey

The three 50-m reaches within each stream were sampled starting > 1 h after sunset (using spotlighting) when kōkopu are active. Sampling consisted of using a high-powered spotlight to scan the reach for fish [41]. This method has been used effectively for sampling kōkopu within wadeable streams at night [42]. Alternative fish sampling methods such as electrofishing and trapping are less effective for surveying kōkopu because these species sink when stunned, occupy deep bank cover during the day, and may not encounter traps due to having high pool fidelity at night [43, 44]. The 1 h delay after sunset ensured that resident kōkopu had left their daytime refuges and moved into nocturnal foraging areas where they could be seen and captured. When spotted, kōkopu generally remained stationary and were able to be caught using

hand nets. When kōkopu were seen but not caught, the estimated length and species of the individual were recorded as a 'miss'. Reaches were sampled using successive depletion passes until fish were no longer observed. This required up to five passes and took around 1.5 h per reach. Captured kōkopu were anaesthetised in 20 mg/L of AQUI-S water-dispersible liquid anaesthetic to facilitate handling. Each fish was identified to species level, measured (total length ± 1 mm), and weighed (± 0.01 g). After measurement, fish were returned to their area of capture. All sampling procedures were approved by the University of Canterbury Animal Ethics Committee (permit number 2020/06R).

## Data analysis

A size class breakpoint of 90 mm (total length) was used to examine how small and large kōkopu responded to habitat characteristics. This breakpoint was selected because banded and giant kōkopu are approximately one year old at this size and begin to compete for territory [45, 46]. Equivalent studies have not been completed for shortjaw kōkopu, but they likely follow a similar pattern and were pooled into the same size class groups for consistency.

Prior to analyses, large giant kōkopu and large shortjaw kōkopu biomass responses were fourth root-transformed, and large banded kōkopu and composite small kōkopu biomass responses were square root-transformed to meet assumptions of normality. One large shortjaw kōkopu biomass outlier was identified using interquartile range criterion and removed. Small size classes were grouped because juvenile shortjaw and giant kōkopu were absent from most reaches due to being naturally rare, which meant we could not develop effective habitat-biomass models for these cohorts. However, this grouping is appropriate because juvenile kōkopu likely occupy the same habitats due to being competitively displaced by larger dominant congeners. Biomass measurements were used as a response instead of counts because kōkopu body mass varies substantially between individuals and is associated with available resources, whereas the association between fish counts and resource availability is also determined by competitive interactions [17]. Variance inflation factors (VIF) were calculated to ensure that there was no collinearity between predictors (i.e., VIF ≤ 4; [47]). Because all VIF values were low (VIF < 2.0), we proceeded with model selection.

To assess the importance of local habitat characteristics in explaining kōkopu biomass, a set of ecologically realistic *a priori* linear mixed-effects models that included all combinations of the four habitat variables was used to explain the biomass of each species' large size class and the composite biomass of small kōkopu size classes. Ecologically realistic interactions between habitat features were initially included, but later removed due to poor data spread creating unreliable results. We originally included the biomass of other kōkopu size classes as fixed effects to test for potentially confounding interspecific interactions, but these were removed because we found no associations between kōkopu of the same size class (S1 Table). Furthermore, any direct effect of kōkopu biomass on the response, particularly between size classes, would likely be indirectly driven by abiotic variables, and we are unable effectively disentangle abiotic and biotic effects using interactions with our dataset. A random factor for stream site, hereafter referred to as 'site', was included so that each of the three 50-m reaches nested within each site could be used independently to examine how habitat characteristics influenced kōkopu biomass. By focusing on the reach-scale, more accurate and informative localised habitat-biomass relationships could be obtained. A random 'catchment' variable was also included to account for any catchment-scale variation shared by sites. Linear mixed-effects models were constructed using the 'lmer' function (Package 'lme4'; [48]) in R version 4.1.1 [49].

An information theoretic approach, using Akaike's information criterion corrected for small sample size ($AIC_c$), was used to determine which candidate models explained variation

in large and small kōkopu biomass most parsimoniously [50]. Assessing model performance using AIC is asymptotically equivalent to using leave-one-cluster-out cross-validation [51]. Each model's $AIC_c$ was subtracted from the lowest $AIC_c$ to determine its $\Delta AIC_c$ [50]. Parsimonious models had $\Delta AIC_c$ values < 2 [50]. Marginal and conditional coefficients of determination ($R^2_m$ and $R^2_c$, respectively) values were calculated for each parsimonious model to evaluate goodness-of-fit because $AIC_c$ only ranks models relative to each other [52]. The marginal coefficient of determination ($R^2_m$) is the proportion of variance explained by fixed effects, whereas the conditional coefficient of determination ($R^2_c$) is the proportion of variance explained by fixed and random effects. The Akaike weight (*w*), interpreted as the probability that a particular model is the most parsimonious model among the candidate models, and $R^2_c$ of parsimonious models were assessed to select the most suitable model for explaining the biomass of each species' composite size classes.

The sum of Akaike weights [53], interpreted as the probability that a predictor is a component of the most parsimonious model, and partial dependence plots were used to assess the relative importance of each habitat variable on the biomass of each kōkopu size class. Partial dependence plots show the independent effect of a single variable on the response by accounting for the average effects of all other variables in a model [54]. Using the 'effects' package [55], partial dependence plots were developed for each species' size class by extracting the independent effects of each variable within a linear mixed-effects model that included all four habitat features and a random factor for site and catchment.

## Results

Large banded kōkopu biomass was explained parsimoniously (i.e., $\Delta AICc$ < 2) by a single model that included pool volume and forest cover ($BK_{LG1}$; Table 1). Pool volume and forest cover were strong explanatory variables, as indicated by relative importance values > 0.75,

**Table 1. Top linear mixed-effects models ($\Delta AIC_c$ < 2) that explain variation in the biomass of large and small banded, giant, and shortjaw kōkopu based on Akaike's information criterion (AIC).**

| Response | Model | Fixed effects | $AIC_c$ | $\Delta AIC_c$ | $w_i$ | $E_r$ | $R^2_m$ | $R^2_c$ |
|---|---|---|---|---|---|---|---|---|
| Large kōkopu | | | | | | | | |
| $\sqrt[2]{\text{Banded kōkopu biomass}}$ | **$BK_{LG1}$** | **PV+FC** | **329.42** | **0.00** | **0.40** | **1.00** | **0.18** | **0.52** |
| $\sqrt[4]{\text{Giant kōkopu biomass}}$ | **$GK_{LG1}$** | **BC+PV** | **196.79** | **0.00** | **0.27** | **1.00** | **0.26** | **0.66** |
| | $GK_{LG2}$ | BC+PV+FC | 197.02 | 0.23 | 0.24 | 1.13 | 0.29 | 0.65 |
| | $GK_{LG3}$ | BC | 198.14 | 1.35 | 0.14 | 1.93 | 0.21 | 0.63 |
| | $GK_{LG4}$ | BC+FC | 198.55 | 1.76 | 0.11 | 2.45 | 0.24 | 0.61 |
| $\sqrt[4]{\text{Shortjaw kōkopu biomass}}$ | **$SJ_{LG1}$** | **SS+PV** | **116.95** | **0.00** | **0.35** | **1.00** | **0.43** | **0.54** |
| | $SJ_{LG2}$ | SS+PV+BC | 117.51 | 0.56 | 0.26 | 1.35 | 0.48 | 0.56 |
| Small kōkopu | | | | | | | | |
| $\sqrt[2]{\text{Composite kōkopu biomass}}$ | **$Composite_{SM1}$** | **SS+BC+PV** | **216.99** | **0.00** | **0.27** | **1.00** | **0.15** | **0.66** |
| | $Composite_{SM2}$ | SS+BC | 218.03 | 1.03 | 0.16 | 1.69 | 0.12 | 0.64 |
| | $Composite_{SM3}$ | SS+BC+PV+FC | 218.67 | 1.67 | 0.12 | 2.25 | 0.15 | 0.63 |

$AIC_c$ represents AIC values corrected for small sample size; delta $AIC_c$ ($\Delta AIC_c$) is the difference in $AIC_c$ score between the highest ranked model and the candidate model; Akaike weight ($w_i$) is the probability that a particular model is the most parsimonious model among the candidate models; evidence ratio ($E_r = w_{top}/w_i$) is the relative comparison in *w* between a candidate model and the top model; marginal coefficient of determination ($R^2_m$); conditional coefficient of determination ($R^2_c$). '$BK_{LG}$' is large banded kōkopu biomass (g), '$GK_{LG}$' is large giant kōkopu biomass (g), '$SJ_{LG}$' is large shortjaw kōkopu biomass (g), '$Composite_{SM}$' is composite small size class biomass, 'PV' is total pool volume ($m^3$), 'FC' is mean forest cover (%), 'BC' is total bank cover (m), and 'SS' is average substrate size (cm) per 50 m reach. Bolded models were selected as the most suitable for predicting biomass within each size class.

whereas substrate size and forest cover poorly explained large banded kōkopu biomass, having relative importance weights < 0.50 (Table 2).

In contrast to banded kōkopu, bank cover was key a predictor in explaining large giant kōkopu biomass, having a relative importance value of 0.97, and featured in each of the four parsimonious models (Tables 1 and 2). The first model ($GK_{LG1}$), which included bank cover and pool volume, was selected as the most suitable because it explained the most variation ($R^2_c$ = 0.66) and was 1.93 times more likely to be the better model than when compared to the simplest model ($GK_{LG3}$), which only included bank cover, as indicated by Akaike weights (0.27/ 0.14; Table 1).

Despite being a poor predictor for other large kōkopu size classes, substrate size was an essential characteristic in explaining large shortjaw kōkopu biomass, having a relative importance value of 1.00, and featured in each of the two parsimonious models (Tables 1 and 2). The first model ($SJ_{LG1}$), which included substrate size and pool volume, was selected as the most suitable because it explained a similar amount of variation in large shortjaw kōkopu biomass to $SJ_{LG2}$, which also included bank cover, despite being simpler ($R^2_c$ ~ 0.55; Table 1). Additionally, $SJ_{LG1}$ was 1.35 times more likely to be the best model, as indicated by Akaike weights (0.35/0.26; Table 1).

Composite small kōkopu biomass was parsimoniously explained by three models, with bank cover and substrate size featuring in each model and receiving relative importance values of 0.88 and 0.68 respectively (Tables 1 and 2). In addition to these influential variables, pool volume featured in the model that most suitably explained small kōkopu biomass (Composite$_{SM1}$; Table 1). Although Composite$_{SM1}$ explained a similar amount of variation to other models ($R^2_c$ ~ 0.65) and included a weak pool volume predictor ($\Sigma w$ = 0.55; Table 2), it was 1.69

**Table 2. Relative importance of habitat features affecting the biomass of large banded kōkopu, giant kōkopu, and shortjaw kōkopu and composite small kōkopu size classes.**

| Response | Fixed effects | $\Sigma w$ |
|---|---|---|
| Large kōkopu | | |
| Banded kōkopu biomass | PV | 0.80 |
| | BC | 0.25 |
| | FC | 0.79 |
| | SS | 0.22 |
| Giant kōkopu biomass | PV | 0.67 |
| | BC | 0.97 |
| | FC | 0.46 |
| | SS | 0.22 |
| Shortjaw kōkopu biomass | PV | 0.79 |
| | BC | 0.49 |
| | FC | 0.23 |
| | SS | 1.00 |
| Small kōkopu | | |
| Composite kōkopu biomass | PV | 0.55 |
| | BC | 0.88 |
| | FC | 0.33 |
| | SS | 0.68 |

Relative variable importance is calculated by summing the Akaike weights ($w$) across the set of models in which the variable appears. Variables exerting a greater influence on kōkopu biomass are characterised by larger summed Akaike weights ($\Sigma w$). Abbreviations for habitat variables are the same as in Table 1.

**Table 3. Summary results of the fixed effects included in the linear mixed-effects models that most parsimoniously predict the biomass of large banded, giant, and shortjaw kōkopu and composite small kōkopu biomass as identified in Table 1.**

| Response | Fixed effects | Coefficient estimate (±SE) |
|---|---|---|
| Large kōkopu | | |
| $\sqrt[2]{\text{Banded kōkopu biomass}}$ | Intercept | 2.16 (±2.19) |
| | PV | 0.08 (±0.03) |
| | FC | 0.06 (±0.03) |
| $\sqrt[4]{\text{Giant kōkopu biomass}}$ | Intercept | -0.06 (±0.47) |
| | BC | 0.04 (±0.01) |
| | PV | 0.02 (±0.01) |
| $\sqrt[4]{\text{Shortjaw kōkopu biomass}}$ | Intercept | -0.41 (±0.22) |
| | SS | 0.10 (±0.02) |
| | PV | 0.01 (±<0.01) |
| Small kōkopu | | |
| $\sqrt[2]{\text{Composite kōkopu biomass}}$ | Intercept | 5.48 (±0.66) |
| | SS | -0.11 (±0.04) |
| | BC | -0.04 (±0.01) |
| | PV | 0.02 (±0.01) |

Abbreviations for habitat variables are the same as in Table 1.

times more likely to be the better model than when compared to the simplest model (Composite$_{SM2}$), which included bank cover and substrate size (Table 1).

Table 3 details the summary statistics for models that most suitably explained the biomass of each species' large size class and composite small size classes (BK$_{LG1}$, GK$_{LG1}$, SJ$_{LG1}$, Composite$_{SM1}$).

In conjunction with relative importance values (Table 2), the effects of habitat characteristics varied greatly between kōkopu species and size classes (Fig 1). With all other variables held constant, pool volume was positively associated with the biomass of large banded kōkopu ($R^2_c$ = 0.20, $F_{1,54.84}$ = 5.99, P = 0.018; Fig 1A), giant kōkopu ($R^2_c$ = 0.14, $F_{1,44.49}$ = 4.14, P = 0.048; Fig 1E), and shortjaw kōkopu ($R^2_c$ = 0.11, $F_{1,55.55}$ = 4.42, P = 0.040; Fig 1I). However, pool volume was not significantly associated with small kōkopu biomass ($R^2_c$ = 0.12, $F_{1,50.48}$ = 3.46, P = 0.069; Fig 1M). Apart from these corresponding associations between pool volume and the biomass of large kōkopu size classes, no other habitat variables were similarly associated between kōkopu cohorts. Bank cover was positively associated with large giant kōkopu biomass ($R^2_c$ = 0.37, $F_{1,45.05}$ = 10.20, P = 0.003; Fig 1F), negatively associated with small kōkopu biomass ($R^2_c$ = 0.35, $F_{1,56.89}$ = 9.47, P = 0.003; Fig 1N), and not associated with the biomass of large banded kōkopu ($R^2_c$ < 0.01, $F_{1,52.62}$ = 0.02, P = 0.886; Fig 1B), or large shortjaw kōkopu ($R^2_c$ = 0.07, $F_{1,50.71}$ = 2.22, P = 0.143; Fig 1J). Large banded kōkopu biomass was the only cohort to be positively associated with forest cover ($R^2_c$ = 0.21, $F_{1,51.51}$ = 5.58, P = 0.02; Fig 1C), whereas there was no association between forest cover and biomass of large giant kōkopu ($R^2_c$ = 0.09, $F_{1,19.15}$ = 2.49, P = 0.131; Fig 1G), large shortjaw kōkopu ($R^2_c$ < 0.01, $F_{1,54.06}$ = 0.13, P = 0.721; Fig 1K), or small kōkopu ($R^2_c$ = 0.04, $F_{1,53.94}$ = 1.13, P = 0.292; Fig 1O). Similarly to bank cover, substrate size had contrasting associations between cohorts, being positively associated with large shortjaw kōkopu biomass ($R^2_c$ = 0.49, $F_{1,55.31}$ = 33.53, P < 0.001; Fig 1L), negatively associated with small kōkopu biomass ($R^2_c$ = 0.23, $F_{1,53.61}$ = 7.04, P = 0.010; Fig 1P), and not associated with the biomass of large banded kōkopu ($R^2_c$ < 0.01, $F_{1,34.39}$ = 0.01, P = 0.94; Fig 1D), or large giant kōkopu ($R^2_c$ < 0.01, $F_{1,6.94}$ = 0.00, P = 0.996; Fig 1H).

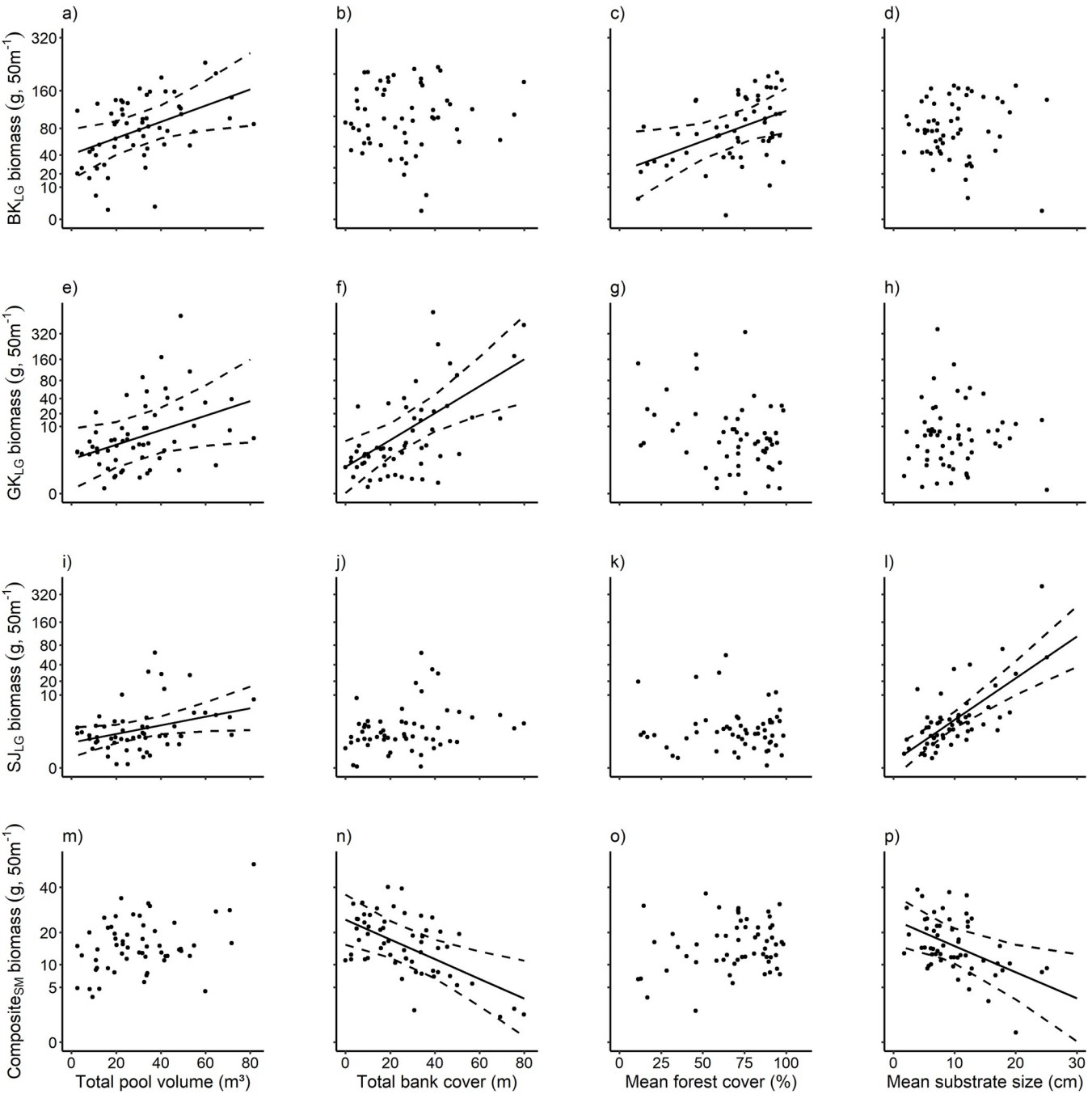

**Fig 1. Independent effects of habitat features on kōkopu biomass.** Partial regression plots of the association between total pool volume, total bank cover, mean forest cover, and mean substrate size and the biomass of large banded kōkopu (BK$_{LG}$; a-d), large giant kōkopu (GK$_{LG}$; e-h), large shortjaw kōkopu (SJ$_{LG}$; i-l), and composite small kōkopu (Composite$_{SM}$; m-p) size classes within each 50-m reach. Note that Y-axes are not linear. Lines of best fit are shown where a significant association was found ($P < 0.05$) and error bands show 95% confidence intervals determined from model fits.

## Discussion

The quantification of body size with respect to specific habitat use is crucial for accurately identifying key habitats that support all life stages of a species and for directing beneficial management and restoration efforts [56]. We identified key habitat features that influence the

biomass of small and large banded, giant, and shortjaw kōkopu, and created statistical models that predict kōkopu biomass based on local habitat features while controlling for other influences. Our results indicate that small and large kōkopu primarily use distinct habitats, and that the influence of habitat characteristics on biomass was particularly variable between the three species' large size classes. By characterising the effects of local habitat characteristics on the biomass of the small and large size classes of each species separately, we provide an accurate and more comprehensive description of kōkopu habitat use.

As hypothesised, total pool volume was a key habitat feature explaining variation in the biomass of each kōkopu species' large size class. Although faster water velocities transport more drifting invertebrates downstream, slower flowing habitats characteristic of pools are commonly associated with greater biomasses of large stream-dwelling fish because they promote greater feeding success through increased strike efficiency and prey capture [14]. However, species like kōkopu and trout still take advantage of greater invertebrate drifts by occupying slow-flowing pools below fast-flowing reaches [5, 8]. Unlike trout, which are predominantly diurnal visual predators, nocturnal galaxiids rely mainly on mechanical lateral line and olfactory sensory systems that work more effectively at slower velocities [57, 58]. Therefore, similarly to other large stream fish, slow-flowing pools are important habitats for large kōkopu, probably because they are profitable foraging areas.

Although pool volume was a key habitat feature for large kōkopu, small kōkopu biomass was only weakly associated with pool habitat, as identified in the AIC analyses, but this association was not significant when all other variables were accounted for. This weak association is likely attributed to small kōkopu occupying the shallow microhabitats on the margins of pools due to being competitively displaced by larger congeners occupying the deeper areas [32, 33]. Small and juvenile fish typically use shallow microhabitats to avoid fast flows, larger predatory fish, or competitively dominant conspecifics [59]. This has been observed in kōkopu, darters (*Percidae* spp.), minnows (*Cyprinidae* spp.), sunfishes (*Centrarchidae* spp.), and salmonids [31, 60, 61]. For example, small cutthroat trout (*Oncorhynchus clarkii*) generally occupy shallow microhabitats and larger adults inhabit deep pools [61]. However, in the absence of large conspecifics, small cutthroats choose large pools and grow more quickly there than in shallow water [18]. Although small kōkopu are likely displaced into less advantageous foraging areas, they still select available habitats with the lowest velocity [31]. This suggests that small kōkopu would likely also use deeper areas of pools if they were not restricted to shallow areas by larger congeners.

In-stream refuges, in addition to pool habitat, were key habitat features that supported large giant and shortjaw kōkopu biomass. Specifically, bank cover was a particularly important predictor for large giant kōkopu, likely because it provides suitable conditions to ambush prey and escape predators [62]. Comparatively, large shortjaw were associated with greater substrate size, which aligns with their affinity for microhabitats with large cobble and boulder substrates that provide interstitial refuge spaces [31, 63]. Refuge use in stream fishes is particularly limited by their body size [60]. This likely explains why the larger-bodied giant kōkopu, which commonly grow to 350 mm (total length), were more limited to larger refuge spaces provided by undercut banks. In-stream refuge is likely also an essential habitat feature for large banded kōkopu, which use a variety of features including boulder interstices, woody debris, and undercut bank cover [25]. This diverse use of refuge types likely explains why we found no clear association with either bank cover or substrate size, because one type of refuge will readily be used when preferred cover is scarce [25]. Therefore, in-stream refuges are key habitat features for large kōkopu, but the type of cover used is likely determined by the type of cover available and individual body size constraints.

Unlike their larger congeners, small kōkopu biomass was negatively associated with bank cover and substrate size. Despite hypothesising that small kōkopu would use these features for

refuges, small kōkopu were likely competitively displaced by larger congeners that have an affinity for these structures. Furthermore, small kōkopu may avoid these habitat structures, which are used by larger opportunistic kōkopu and predatory longfin eels (*Anguilla dieffenbachii*) that ambush prey [64, 65]. Size-related refuge use can also be dependent on the densities of adults. For example, smaller European bullhead (*Cottus gobio*) juveniles used microhabitats with larger substrate alongside adults when fish densities were low, but were displaced into microhabitats with smaller substrate when fish densities increased [66]. Therefore, similarly to pool habitat use, it appears that available in-stream cover would likely be utilised more by small kōkopu when predators and competitively dominant congeners are absent, but these potential refuges are avoided, and shallow microhabitats are used instead, when these larger fish are present.

Unlike other habitat features that had multiple associations with kōkopu cohorts, forest cover was solely associated with greater large banded kōkopu biomass. This was unexpected because forested streams generally provide important terrestrially-derived food subsidies that can support greater fish biomass and contribute up to half the annual energy budget of some drift-feeding species [11, 67]. The observed interspecific variability could be attributed to the relative importance of terrestrial invertebrates in the diet of the three kōkopu species. Terrestrial invertebrates are an essential food resource for banded kōkopu, making up around 75% numerically and 90% gravimetrically of their diet, and this may explain their close association with greater forest cover [44]. In comparison, terrestrial invertebrates make up around 42% numerically and 48% gravimetrically of giant kōkopu diet, and 23% numerically and 51% gravimetrically of shortjaw kōkopu diet, indicating these species may have a relatively reduced reliance on terrestrial food subsidies [26, 68]. Furthermore, the absence of positive relationships with the rarer large shortjaw and giant kōkopu could be driven by the more abundant banded kōkopu likely intercepting most available food. Similarly, small kōkopu likely had no association with forest cover because they are competitively displaced from key feeding areas and cannot access terrestrial subsidies [17]. It is important to note that terrestrial food resources may also be sourced from forested areas upstream of the study reach or from low-hanging riparian vegetation that our densiometer measurements did not include. Therefore, the relationships between kōkopu cohorts and local forest cover were likely driven by a combination of competition, food availably, and prey preference.

Overall, our results indicate that there are variable associations between local habitat characteristics and the biomass of kōkopu species and their composite size classes. This indicates that habitat restoration efforts should consider the habitat use of each kōkopu size class concurrently. Despite conspecific and congeneric conflicts in habitat-biomass associations, it is important to identify habitat characteristics that provide the greatest benefits to the population of reproductively viable adults [69]. If juvenile habitats are limited or degraded, adult populations may become recruit-limited [70]. However, if an adult population typically has excess recruits, and is limited by habitat, then the most beneficial management decisions would prioritise adult habitat. Often, a balance of adult and juvenile habitat requirements must be incorporated into management restoration to benefit the population overall. For example, gravel augmentation is a key tool used to restore salmonid spawning and incubation grounds, but conflicts arise when adult Chinook salmon (*Oncorhynchus tshawytscha*) preferentially spawn in fine gravels where embryo survival is least likely [71]. Therefore, intermediate-sized gravels would likely maximise overall reproductive success across both spawning and incubation life stages. Similar trade-offs need to be considered in kōkopu management to balance the habitat needs of juveniles and adults and to provide the greatest net benefit to kōkopu populations overall.

The advantages of our localised models over pre-existing New Zealand fish distribution models (e.g., [72–75]) is that they (1) provide standardised biomass estimations for small and large size classes instead of species presence; (2) were developed using consistent methodology at an explicitly local spatial scale; and (3) avoided confounding influences by only including streams with no downstream barriers, invasive species, or notable degradation. The accuracy and transferability of our models could be improved by sampling a wider array of streams with different kōkopu compositions, allowing abiotic and biotic effects to be disentangled and small giant and shortjaw kōkopu size classes to be included. Regardless, these standardised predictions are still applicable to kōkopu populations across New Zealand because we sampled reaches with an array of species compositions that are naturally realistic and common, thus providing a robust description of the average relationship between species' biomass and their local environment (S2 and S3 Tables). It is important to note that these patterns may vary regionally, particularly when applied to areas less accessible to kōkopu or with more impacted catchments. Therefore, our models will work most effectively when other factors are accounted for and can only be applied to reaches that are within the natural distribution of the target kōkopu species. Patterns may also vary seasonally due to small kōkopu immigration events in spring, and the subsequent gain and loss of large kokopu biomass associated with spawning in autumn. It is unlikely that known seasonal shifts in kōkopu microhabitat use (i.e., position within a pool) will influence our local habitat associations substantially because fish generally remain within the same relatively short home range [76]. Further long-term monitoring across New Zealand is required to confirm these relationships.

In conclusion, our study demonstrates the importance of examining size-related habitat use when identifying key habitats that support species and provides an effective modelling approach for predicting the biomass of small and large-size classes of stream fish using simple habitat measurements. This enhanced understanding of how kōkopu size classes are influenced by their local environments allows a standardised prediction and baseline of likely kōkopu biomass based on local habitat characteristics within minimally degraded coastal catchments. These standardised predictions could be used to isolate and accurately test the immediate effects of anthropogenic stressors on local populations of these declining endemic species [69]. Through effective evaluation of population densities, guiding habitat restoration efforts, and helping direct actions to mitigate anthropogenic stressors [77], the types of modelling techniques used here will be a useful tool for conserving freshwater fish.

## Supporting information

**S1 Table. Summary statistics of the effects of each kōkopu species' biomass on the biomass of other kōkopu within the same size class, after accounting for habitat features.**
(DOCX)

**S2 Table. Instances of specific co-occurrence between banded (BK), giant (GK), and shortjaw (SJ) kōkopu small (SM) and large (LG) size classes observed across 57 sampled reaches.**
(DOCX)

**S3 Table. Instances of general co-occurrence between banded (BK), giant (GK), and shortjaw (SJ) kōkopu small (SM) and large (LG) size classes observed across 57 sampled reaches.**
(DOCX)

## Acknowledgments

We are grateful to Chris Meijer, Anne Gagné, and the Marine Ecology Research Group for extensive field assistance; Spencer Virgin and Helen Warburton for insightful analytical

assistance; the Freshwater Ecology Research Group for discussion and feedback; and two anonymous reviewers for valuable comments that greatly improved earlier versions of this manuscript. We would also like to thank the New Zealand Ministry of Business, Innovation and Employment, Department of Conservation, and University of Canterbury for financial and logistical support.

## Author Contributions

**Conceptualization:** Ben R. J. Crichton, Michael J. H. Hickford, Angus R. McIntosh, David R. Schiel.

**Data curation:** Ben R. J. Crichton, Michael J. H. Hickford, Angus R. McIntosh, David R. Schiel.

**Formal analysis:** Ben R. J. Crichton.

**Funding acquisition:** Michael J. H. Hickford, David R. Schiel.

**Investigation:** Ben R. J. Crichton, Michael J. H. Hickford, Angus R. McIntosh, David R. Schiel.

**Methodology:** Ben R. J. Crichton, Michael J. H. Hickford, Angus R. McIntosh, David R. Schiel.

**Project administration:** Ben R. J. Crichton, Michael J. H. Hickford, David R. Schiel.

**Resources:** Michael J. H. Hickford, David R. Schiel.

**Supervision:** Michael J. H. Hickford, Angus R. McIntosh, David R. Schiel.

**Validation:** Ben R. J. Crichton, Michael J. H. Hickford, Angus R. McIntosh, David R. Schiel.

**Visualization:** Ben R. J. Crichton, Michael J. H. Hickford, Angus R. McIntosh, David R. Schiel.

**Writing – original draft:** Ben R. J. Crichton.

**Writing – review & editing:** Ben R. J. Crichton, Michael J. H. Hickford, Angus R. McIntosh, David R. Schiel.

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
