## [Decision Letter · Decision Letter 0]

22 Feb 2022

PONE-D-21-39491Predicting biomass of resident kōkopu (*Galaxias*) populations using local habitat compositionPLOS ONE

Dear Dr. Crichton,

Thank you for submitting your manuscript to PLOS ONE. After careful consideration, we feel that it has merit but does not fully meet PLOS ONE’s publication criteria as it currently stands. Therefore, we invite you to submit a revised version of the manuscript that addresses the points raised during the review process.I have now received two reviews of your manuscript. Your manuscript got many general but also quite some specific comments that you need to consider before we move forward in the publication process. I found that the potential of this manuscript critically depends on how your arguments can meet the reviewers comments. A major concern raised by both reviewers was that species identity was not included in the models. Although there is a large overlap in microhabitat it is still plausible that the different species have different ecological requirements. Both reviewers have given suggestions on how this can be dealt with. Another concern regards the representativeness of your results both to other systems and rivers entailing kökopu populations but also to other freshwater species. In your response letter you need to include point by point how you dealt with the reviewers' comments.

We look forward to receiving your revised manuscript.

Kind regards,

Peter Eklöv

Academic Editor

PLOS ONE

Journal Requirements:

4. Please include a caption for figure 1

Reviewers' comments:

Reviewer's Responses to Questions

**Comments to the Author**

1. Is the manuscript technically sound, and do the data support the conclusions?

Reviewer #1: Yes

Reviewer #2: Partly

2. Has the statistical analysis been performed appropriately and rigorously? 

Reviewer #1: Yes

Reviewer #2: No

3. Have the authors made all data underlying the findings in their manuscript fully available?

Reviewer #1: Yes

Reviewer #2: No

4. Is the manuscript presented in an intelligible fashion and written in standard English?

Reviewer #1: Yes

Reviewer #2: Yes

5. Review Comments to the Author

Reviewer #1: The manuscript “Predicting biomass of resident kōkopu (Galaxias) populations using local habitat composition” evaluates the relationship among biomass of three endemic species of Galaxias and five instream habitat characteristics in New Zealand streams. I think the study is very well conducted and well written, and it is of great contribution to the field. This research is not only important for conservation of the species under study, but can also be used as a model to understand what drives variation in stream fish populations elsewhere. In general, I have mostly minor comments to the manuscript. My main concern is about the not inclusion of species identity in the models; all the three species are regarded as a single population in the models, which may be an issue because one may expect at least little differences in microhabitat selection. I think that adding the species identity as a random variable in the models will make the study more compelling and convincing to the readers.

Title: I would suggest to use “local habitat characteristics' ' instead “composition”, as composition is more related to the content (like for a chemical compound or for a fish assemblage).

Abstracts: Is it a riverine or stream fish? Or a lake? Maybe state the type of freshwater habitat.

Introduction:

L60-74: Regarding this, you may want to check Dala-Corte & De Fries (2018) study, which investigates very similar questions.

Dala-Corte, R. B., & De Fries, L. (2018). Inter and intraspecific variation in fish body size constrains microhabitat use in a subtropical drainage. Environmental Biology of Fishes, 101(7), 1205-1217.

L66. Give an example for what you mean with “same local environment”.

L119: “habitat characteristics”. Review it elsewhere in the text.

L119: State again which are these 3 species.

L119-131: This paragraph is almost entirely a methods paragraph, I would move to methods.

L133: I recommend “kokopu fish populations of three species endemic to New Zealand streams”.

L133: There is no need to state the use of AIC here, leave it to methods only. AIC is a widely known statistical approach.

L136-142: Congratulations for the predictions.

Methods

L173-193: Were the individuals identified to species-level during the field?

Data analysis

L195: I think the author should include the species identity in the models (at least as a random variable). I know that the three species may largely overlap their microhabitat selection, readers will still think that it is very plausible that the different species have slightly different ecological requirements and behaviors. For instance, see Dala-Corte and De Fries (2018).

Results

Table 1. Tables should not have these horizontal lines in the middle.

L246: Like you did with large kokopu, start the small kokopu paragraph stating the results for the AIC models.

L256-258: Actually, looking at the deltaAICc (smaller than 2), you do not have strong support to say that the model 1S is better than the 4S. So theoretically you should say that the best model is the 4S, which is the most simple. Yet, it is important to discuss the model 1S too.

Table 2. Since you may be interested in the effect size of all the explanatory variables, you may want to inform the relative importance of the variables using the Summed Akaike Weights (SW) of all the models. See Burnham and Anderson (2002) and Giam and Olden (2016) if you have interest.

Burnham KP, Anderson DR (2002) Model selection and multimodel inference: a practical information-theoretic approach. Springer, New York.

Giam X, Olden JD (2016) Quantifying variable importance in a multi-model inference framework. Methods Ecol Evol 7:388–397

Discussion

You may want to expand your comparisons to other studies with stream fish besides those with kokopu and trouts. For example, see Dala-Corte and De Fries (2018).

L283-292: You may want to recall the readers who are the kokopu species you studied.

L347. Ok, it iwas not locally important, but give an emphasis in this conclusion that upstream forest cover may be essential and probably explain your results.

L357-359: Ok, so benthic insects may be an important feeding resource. Make it clearer.

L348-365. Indeed, I think that local riparian forest may be more important in driving refuge for small fish than in predicting the local input of terrestrial invertebrates, because upstream riparian forest may provide plenty of drifting terrestrial invertebrates to the pools.

L367-373. You may want to take into consideration that the negative relationship between total bank cover and small kokopu biomass may just reflect that large kokopus displace the small ones from sites with more bank cover. In other words, an increase in bank cover may not be a conflict of interests for habitat restoration between large and small kokopus. If streams had no bank cover, you could probably don’t see any kokopu at all. To assess this one must sampling long stream reaches with few or no bank cover at all.

Reviewer #2: Dear Authors,

In my view, the basic premise of this manuscript – the need for better understanding and characterisation of ontogenetic variations in physical habitat preferences – is sound. However, I feel that there are a few aspects of this study that need to be tackled more substantively before it is ready for publication.

Conceptually, I see the value in pursuing the development of predictive models of fish biomass to support species management. As such, I see this as a valuable endpoint for this study. However, to strengthen the credibility and hence uptake of such models, in my view there needs to be greater transparency around the underlying data and assumptions that have been made. Furthermore, there is a need for validation and consideration of the transferability of the resulting models.

The study offers a valuable dataset on kōkopu habitat utilisation from multiple rivers on the west coast of the South Island of New Zealand. It is predicated on an assumption of potential habitat separation between ‘small’ and ‘large’ individuals arising from competitive displacement of smaller individuals from preferred habitats as indicated by previous studies. Below I make a few observations regarding the study methodology and some of the assumptions that I feel need addressing within the manuscript.

Firstly, it is indicated that the fish surveys took place during May and June (L147). This coincides with the spawning season of these species, yet the potential implications of this in terms of fish behaviour and habitat choice (and hence representativeness/transferability of these data and models) are not addressed in the manuscript. Mature individuals are known to aggregate prior to spawning and can undertake spawning migrations. Consequently, how representative are these observations (particularly for the ‘large’ group) of ‘normal’ habitat use and hence how transferable are the models?

Secondly, observations for the three kōkopu species have been pooled, based on an assumption that they regularly co-exist and have similar habitat requirements (L121). Broadly speaking, yes, they all typically display an association with pool habitats and some degree of overlap in distribution does occur. However, I would suggest that there’s sufficient evidence to indicate a degree of spatial separation in the distribution of the three species (particularly beyond the South Island west coast rivers) broadly correlating with distance inland and elevation (giants lower gradient closer to the sea, bandeds low-mid gradients and further inland, shortjaws mid-higher gradients further inland still). I would also think there is some potential for differentiation in micro-habitat use between the species. Consequently, I feel there is a need to account for this potential in the analyses or at least demonstrate the assumption is sound. One of the benefits of the more modern mixed models as you have used here is that it would be simple to include species as a predictor variable and check the validity of this assumption. You show no data on the proportional make up of your observations between the species, nor information on equivalence (or not) of size classes between the species. Likewise, there is no exploration of whether there is any spatial or microhabitat separation between the species (e.g. do shortjaws inhabit larger pools than bandeds?). Much of this would be fairly straightforward to illustrate graphically. With respect to species management, these factors are crucial to understand and be confident about for ensuring the validity and credibility of your models for informing management actions.

Thirdly, and somewhat linked to the above, how representative are the west coast river systems and kōkopu populations of other rivers and kōkopu populations across New Zealand, and hence, how transferable are the results to elsewhere in the country?

Fourthly, is there an interaction between pool size and pool depth? Intuitively I would expect the potential for some statistical redundancy between these two variables. It would be good to see this addressed. Likewise, how about between any of the other predictor variables? How closely correlated are they? Similarly, does the apparent correlation between smaller individuals and forest cover simply reflect a preference for smaller streams (and hence greater chance of canopy closure)?

Finally, and this is in my view a particularly critical omission, there is no cross-validation of the statistical models. Good practice in this sort of study would typically see some form of model validation undertaken and reported. For example, using a proportion of the dataset to train the model and a separate portion of the dataset to test and quantify model predictive performance. The addition of this sort of approach would help to provide possible end users greater confidence in the relevance and transferability of the model for real world application.

More generally, I felt that the introduction was biased towards pelagic, drift-feeding species, without really acknowledging that some of the statements regarding energetic benefits of pool feeding etc may not be as generalisable as could be inferred. You also jump into some fairly bold statements about statistical models in your opening paragraph (e.g. L42) that in my view aren’t as well justified as they could be and perhaps need a bit more introductory context.

One specific comment about your choice of sampling for determining mean pool depth – why only measure depths along one axis of the pool? Can you be sure that is representative of the mean pool depth? Do you have a reference for this method ahead of others (e.g. 10 random depth measurements or 5 on each axis)? I would have thought there was quite a bit of depth variation in the longitudinal axis.

In the discussion (and throughout), I think it would be good to consider your terminology with respect to habitat use/requirements/preference. These three terms have different meanings to me – habitat use being the habitats that fish are using (i.e. where you find them when you look for them), habitat preference being where they would most like to be if the habitat was available, and habitat requirements being habitats that are required to fulfil a particular function. These can all be the same, but could also all be distinct. You talk, for example, about small and large kōkopu having different habitat ‘requirements’ (e.g. L288). Do they really require different habitats? Or do they, as you allude to later, have overlapping fundamental niches but different realised niches (L317-318) due to competition for ‘preferred’ space? Or perhaps there is a genuine spatial segregation of small and large kōkopu habitats, with the distribution of smaller fish skewed towards smaller, lower order streams, and larger fish in larger, main stem streams, that may be more indicative of differing habitat preferences/requirements? I think this could be explored in more depth through your analysis, for example by illustrating the random effects of site to show how these relationships may vary by stream size. I know this may seem somewhat pedantic, but there are crucial distinctions in these terms and what they may mean from a management/restoration perspective, so it’s important to be precise about what you mean.

I feel there is real promise in this manuscript, and it is generally well written, but as identified above I do feel there is still some work to achieve the impact that befits the hard work that has gone into collecting these data.

Best of luck with the manuscript.

6. PLOS authors have the option to publish the peer review history of their article (what does this mean?). If published, this will include your full peer review and any attached files.

Reviewer #1: No

Reviewer #2: No

---

## [Author Response · Author response to Decision Letter 0]

19 May 2022

Response to Reviewers

Please note that the line numbers in responses refer to the clean manuscript copy.

Editor:

• This has been corrected.

We note that the grant information you provided in the ‘Funding Information’ and ‘Financial Disclosure’ sections do not match. When you resubmit, please ensure that you provide the correct grant numbers for the awards you received for your study in the ‘Funding Information’ section.

• This has been corrected.

We note that you have stated that you will provide repository information for your data at acceptance. Should your manuscript be accepted for publication, we will hold it until you provide the relevant accession numbers or DOIs necessary to access your data. If you wish to make changes to your Data Availability statement, please describe these changes in your cover letter and we will update your Data Availability statement to reflect the information you provide.

• We have created and linked a Figshare folder that includes the dataset.

link: https://figshare.com/s/686b54046be999b5ed95

DOI: 10.6084/m9.figshare.17203961

Please include a caption for figure 1

• This was given in-text. It is located between L305-311.

Reviewer #1: 

The manuscript “Predicting biomass of resident kōkopu (Galaxias) populations using local habitat composition” evaluates the relationship among biomass of three endemic species of Galaxias and five instream habitat characteristics in New Zealand streams. I think the study is very well conducted and well written, and it is of great contribution to the field. This research is not only important for conservation of the species under study, but can also be used as a model to understand what drives variation in stream fish populations elsewhere. In general, I have mostly minor comments to the manuscript. My main concern is about the not inclusion of species identity in the models; all the three species are regarded as a single population in the models, which may be an issue because one may expect at least little differences in microhabitat selection. I think that adding the species identity as a random variable in the models will make the study more compelling and convincing to the readers.

I would suggest to use “local habitat characteristics' ' instead “composition”, as composition is more related to the content (like for a chemical compound or for a fish assemblage).

• L1: Swapped out “composition” for “characteristics”.

Is it a riverine or stream fish? Or a lake? Maybe state the type of freshwater habitat.

• L31: Described that stream reaches were being sampled.

L60-74: Regarding this, you may want to check Dala-Corte & De Fries (2018) study, which investigates very similar questions.

• Dala-Corte & De Fries (2018) was a great resource and its findings have been addressed between L70-73. The discussion has also benefitted from its themes.

L66. Give an example for what you mean with “same local environment”.

• Have added “(eg., the same stream reach)” (L74).

L119: “habitat characteristics”. Review it elsewhere in the text.

• Have fixed this issue throughout.

L119: State again which are these 3 species.

• L126: Added species names.

L119-131: This paragraph is almost entirely a methods paragraph, I would move to methods.

• This paragraph has been moved to methods (L192-197).

L133: I recommend “kokopu fish populations of three species endemic to New Zealand streams”.

• Sentence has been changed, but similar detail has been added (L126).

L133: There is no need to state the use of AIC here, leave it to methods only. AIC is a widely known statistical approach.

• Removed mentioning of AIC.

L136-142: Congratulations for the predictions.

• Thank you, I am glad we now have even cooler ones to show.

L173-193: Were the individuals identified to species-level during the field?

• Each fish was identified to species level (added L186).

L195: I think the author should include the species identity in the models (at least as a random variable). I know that the three species may largely overlap their microhabitat selection, readers will still think that it is very plausible that the different species have slightly different ecological requirements and behaviors. For instance, see Dala-Corte and De Fries (2018).

• Analyses now focus on each species separately. The biomass of other kōkopu species of the equivalent size-class not being examined have been added as random variables so we can account for species interactions and obtain a better representation of each species’ habitat use (L220-223). The new addition of variance components (in Table 1) indicates how much random variation the other species account for.

Table 1. Tables should not have these horizontal lines in the middle.

• Lines were removed and replaced with shading.

L246: Like you did with large kokopu, start the small kokopu paragraph stating the results for the AIC models.

• Results paragraph structuring is now more consistent between size-classes and mentions AIC information first.

L256-258: Actually, looking at the deltaAICc (smaller than 2), you do not have strong support to say that the model 1S is better than the 4S. So theoretically you should say that the best model is the 4S, which is the most simple. Yet, it is important to discuss the model 1S too.

• This exact case is no longer included, but a similar case has occurred with the small banded kōkopu models. Although BKS2 was a simpler model, BKS1 was selected because it had a notably higher AICwt, indicating it was more likely to be the most parsimonious model. This is explained between L315-319.

Since you may be interested in the effect size of all the explanatory variables, you may want to inform the relative importance of the variables using the Summed Akaike Weights (SW) of all the models. See Burnham and Anderson (2002) and Giam and Olden (2016) if you have interest.

• This was a great addition and is now shown in Table 3.

You may want to expand your comparisons to other studies with stream fish besides those with kokopu and trouts. For example, see Dala-Corte and De Fries (2018).

• Have now included studies using other freshwater fish in citations throughout and in written examples (e.g., L371-374 & L405-408).

L283-292: You may want to recall the readers who are the kokopu species you studied.

• Species have been added (L347).

L347. Ok, it was not locally important, but give an emphasis in this conclusion that upstream forest cover may be essential and probably explain your results.

• L412-430: Paragraph has been changed due to different results. Paragraph now discusses and concludes that the relationships between local forest cover measurements and the biomass of large kōkopu was likely driven by food availably and interspecific competition.

L357-359: Ok, so benthic insects may be an important feeding resource. Make it clearer.

• L440-443: Sentence has been made clearer.

L348-365. Indeed, I think that local riparian forest may be more important in driving refuge for small fish than in predicting the local input of terrestrial invertebrates, because upstream riparian forest may provide plenty of drifting terrestrial invertebrates to the pools.

• L431-450: We believe refuge and aquatic prey are more important drivers of greater small kōkopu biomass because terrestrial prey are likely dominated by larger fish in the pools.

L367-373. You may want to take into consideration that the negative relationship between total bank cover and small kokopu biomass may just reflect that large kokopus displace the small ones from sites with more bank cover. In other words, an increase in bank cover may not be a conflict of interests for habitat restoration between large and small kokopus. If streams had no bank cover, you could probably don’t see any kokopu at all. To assess this one must sampling long stream reaches with few or no bank cover at all.

• Acknowledged that our models describe the realised niche of each species’ size-class between L479-483. This is also emphasised in the paragraphs discussing the relationships between small kōkopu biomass and pool habitats (L378-380) and refuge use (L408-411).

Reviewer #2: 

Dear Authors,

In my view, the basic premise of this manuscript – the need for better understanding and characterisation of ontogenetic variations in physical habitat preferences – is sound. However, I feel that there are a few aspects of this study that need to be tackled more substantively before it is ready for publication.

Conceptually, I see the value in pursuing the development of predictive models of fish biomass to support species management. As such, I see this as a valuable endpoint for this study. However, to strengthen the credibility and hence uptake of such models, in my view there needs to be greater transparency around the underlying data and assumptions that have been made. Furthermore, there is a need for validation and consideration of the transferability of the resulting models.

The study offers a valuable dataset on kōkopu habitat utilisation from multiple rivers on the west coast of the South Island of New Zealand. It is predicated on an assumption of potential habitat separation between ‘small’ and ‘large’ individuals arising from competitive displacement of smaller individuals from preferred habitats as indicated by previous studies. Below I make a few observations regarding the study methodology and some of the assumptions that I feel need addressing within the manuscript.

Firstly, it is indicated that the fish surveys took place during May and June (L147). This coincides with the spawning season of these species, yet the potential implications of this in terms of fish behaviour and habitat choice (and hence representativeness/transferability of these data and models) are not addressed in the manuscript. Mature individuals are known to aggregate prior to spawning and can undertake spawning migrations. Consequently, how representative are these observations (particularly for the ‘large’ group) of ‘normal’ habitat use and hence how transferable are the models?

• Addressed between L141-144. We believe the representativeness of our findings are not influenced by kōkopu spawning because kōkopu in our study were not notably fecund (examined using light stripping action) and because spawning occurs on the banks adjacent to adult habitats.

Secondly, observations for the three kōkopu species have been pooled, based on an assumption that they regularly co-exist and have similar habitat requirements (L121). Broadly speaking, yes, they all typically display an association with pool habitats and some degree of overlap in distribution does occur. However, I would suggest that there’s sufficient evidence to indicate a degree of spatial separation in the distribution of the three species (particularly beyond the South Island west coast rivers) broadly correlating with distance inland and elevation (giants lower gradient closer to the sea, bandeds low-mid gradients and further inland, shortjaws mid-higher gradients further inland still). I would also think there is some potential for differentiation in micro-habitat use between the species. Consequently, I feel there is a need to account for this potential in the analyses or at least demonstrate the assumption is sound. One of the benefits of the more modern mixed models as you have used here is that it would be simple to include species as a predictor variable and check the validity of this assumption. 

• Analyses now focus on each species separately. The biomass of other kōkopu species of the equivalent size-class not being examined have been added as random variables so we can account for species interactions and obtain a better representation of each species’ habitat use (L220-223). The new addition of variance components (in Table 1) indicates how much random variation the other species account for.

• As now mentioned between L479-483, the random effect for stream site should account for variation caused by stream order, distance inland, and elevation that may also drive different species’ habitat use.

• In regard to microhabitat use, please see below comment.

You show no data on the proportional make up of your observations between the species, nor information on equivalence (or not) of size classes between the species. Likewise, there is no exploration of whether there is any spatial or microhabitat separation between the species (e.g. do shortjaws inhabit larger pools than bandeds?). Much of this would be fairly straightforward to illustrate graphically. With respect to species management, these factors are crucial to understand and be confident about for ensuring the validity and credibility of your models for informing management actions.

• Figure 1 now shows the biomass of each species. Because our habitat and fish sampling focused on the 50 m reach scale, we cannot accurately explore microhabitat relationships (e.g., at the pool scale). Attempts to include variables that could potentially be used to examine more microhabitat scales such as average pool volume were unsuitable. For example, reaches could have one large pool and one very small pool, which produced a highly variable average pool size that would not be accurately linked to fish biomass. Nevertheless, current analyses at the reach scale provide practical biomass estimates for each species.

Thirdly, and somewhat linked to the above, how representative are the west coast river systems and kōkopu populations of other rivers and kōkopu populations across New Zealand, and hence, how transferable are the results to elsewhere in the country?

• West Coast river systems and catchments are not notably different from other New Zealand regions that contain kōkopu. Additionally, our standardised predictions should be applicable to kōkopu populations across New Zealand because they describe the realised niche of each size-class and because the random site factor accounts for variation caused by stream order, distance inland, elevation, and disturbance events that often alter kōkopu species’ densities (L479-483).

Fourthly, is there an interaction between pool size and pool depth? Intuitively I would expect the potential for some statistical redundancy between these two variables. It would be good to see this addressed. Likewise, how about between any of the other predictor variables? How closely correlated are they? Similarly, does the apparent correlation between smaller individuals and forest cover simply reflect a preference for smaller streams (and hence greater chance of canopy closure)?

• Pool area and depth have been combined and replaced with pool volume, which better represents available habitat space. As indicated between L206-208, multicollinearity was checked using variance inflation factors and were notably low (<2).

• Unfortunately, the wetted width of the streams were not consistently measured. However, any correlation between wetted width and forest cover would likely be minimised due to the wide viewing angle of the convex densiometer used and because all streams were between 2-6 m wide. Furthermore, we would expect that pool volume and wetted width would be correlated due to larger streams likely having larger pools, but due to low VIF values (<2) between pool volume and forest cover, this indicates that is unlikely that forest cover was correlated with wetted width.

Finally, and this is in my view a particularly critical omission, there is no cross-validation of the statistical models. Good practice in this sort of study would typically see some form of model validation undertaken and reported. For example, using a proportion of the dataset to train the model and a separate portion of the dataset to test and quantify model predictive performance. The addition of this sort of approach would help to provide possible end users greater confidence in the relevance and transferability of the model for real world application.

• L226-227: We take your point and did consider it. However, according to Fang (2011), further cross validation analyses are unnecessary because we used maximum likelihood estimates within our AIC analyses that are asymptotically equivalent to the leave-one-cluster-out cross-validation.

More generally, I felt that the introduction was biased towards pelagic, drift-feeding species, without really acknowledging that some of the statements regarding energetic benefits of pool feeding etc may not be as generalisable as could be inferred. You also jump into some fairly bold statements about statistical models in your opening paragraph (e.g. L42) that in my view aren’t as well justified as they could be and perhaps need a bit more introductory context.

• Reworded sentence (L48-50) to be less bold. Remaining sentences seem suitable. 

• Added “drift-feeding” to pool description (L57) because it applies more to drift-feeding fish. However, statements about in-stream cover, forest canopy cover, and size-related hierarchies have been left generally applicable because they apply to most stream fish, whether pelagic or demersal.

One specific comment about your choice of sampling for determining mean pool depth – why only measure depths along one axis of the pool? Can you be sure that is representative of the mean pool depth? Do you have a reference for this method ahead of others (e.g. 10 random depth measurements or 5 on each axis)? I would have thought there was quite a bit of depth variation in the longitudinal axis.

• Measurements were taken from the width axis because measurements taken from the length axis were presumed to be biased by following the thalweg, which remained rather consistently deep along the pools. Our method has been noted in the text (L166-167), but we acknowledge this could have been completed more effectively (i.e., take measurements across the width at the beginning, middle, and end of the pool). We do not feel that this will notably alter the accuracy of our pool volume results.

In the discussion (and throughout), I think it would be good to consider your terminology with respect to habitat use/requirements/preference. These three terms have different meanings to me – habitat use being the habitats that fish are using (i.e. where you find them when you look for them), habitat preference being where they would most like to be if the habitat was available, and habitat requirements being habitats that are required to fulfil a particular function. These can all be the same, but could also all be distinct. You talk, for example, about small and large kōkopu having different habitat ‘requirements’ (e.g. L288). Do they really require different habitats? Or do they, as you allude to later, have overlapping fundamental niches but different realised niches (L317-318) due to competition for ‘preferred’ space? Or perhaps there is a genuine spatial segregation of small and large kōkopu habitats, with the distribution of smaller fish skewed towards smaller, lower order streams, and larger fish in larger, main stem streams, that may be more indicative of differing habitat preferences/requirements? I think this could be explored in more depth through your analysis, for example by illustrating the random effects of site to show how these relationships may vary by stream size. I know this may seem somewhat pedantic, but there are crucial distinctions in these terms and what they may mean from a management/restoration perspective, so it’s important to be precise about what you mean.

• Terminology has been fixed to reflect our examination of habitat use. As explained in an above comment, stream width was not measured. However, stream related variables like elevation, distance inland, and stream order should be accounted for by the random ‘site’ variable. The variation accounted for by site is now shown in Table 1.

---

## [Decision Letter · Decision Letter 1]

23 Jun 2022

PONE-D-21-39491R1Predicting biomass of resident kōkopu (*Galaxias*) populations using local habitat characteristicsPLOS ONE

Dear Dr. Crichton,

Thank you for submitting your manuscript to PLOS ONE. After careful consideration, we feel that it has merit but does not fully meet PLOS ONE’s publication criteria as it currently stands. Therefore, we invite you to submit a revised version of the manuscript that addresses the points raised during the review process. I have received a second review from one of the reviewers and as you can this reviewer is happy with the development of this manuscript after the first round of review. However, the reviewer has also some additional comments that need to be resolved before we can proceed in the publication process. Please, indicate in your response letter, point by point, how you have dealt with this reviewer's comments.

We look forward to receiving your revised manuscript.

Kind regards,

Peter Eklöv

Academic Editor

PLOS ONE

Reviewers' comments:

Reviewer's Responses to Questions

**Comments to the Author**

1. If the authors have adequately addressed your comments raised in a previous round of review and you feel that this manuscript is now acceptable for publication, you may indicate that here to bypass the “Comments to the Author” section, enter your conflict of interest statement in the “Confidential to Editor” section, and submit your "Accept" recommendation.

Reviewer #2: (No Response)

2. Is the manuscript technically sound, and do the data support the conclusions?

Reviewer #2: Partly

3. Has the statistical analysis been performed appropriately and rigorously? 

Reviewer #2: No

4. Have the authors made all data underlying the findings in their manuscript fully available?

Reviewer #2: Yes

5. Is the manuscript presented in an intelligible fashion and written in standard English?

Reviewer #2: Yes

6. Review Comments to the Author

Reviewer #2: Dear Authors,

The changes made to the manuscript in response to the initial reviews are positive and validate the comments from both reviewers regarding the need to account for the separate species in your models.

The basic premise of this manuscript remains sound, and it is based on a strong empirical foundation. The species-specific models have improved the value and credibility of this work, but in my view there remain some questions regarding the structure of your statistical models and, potentially, their transferability that need to be addressed.

Firstly, in my view it is not appropriate to include the biomass of other species as random effect – this should be treated as a fixed effect because, as you state, it is included to account for the direct effect of competitive exclusion.

Secondly, it appears that you are treating all sampled reaches as independent sites, even though they are nested within catchments and hence subject to correlated effects. In my view this is potentially an inappropriate treatment of pseudo-replication and the spatial structuring of reaches by catchment needs to be accounted for in your formulation of your site random effect (e.g. reaches need to be nested within catchments).

Thirdly, I am not yet sufficiently convinced of the transferability of your models to areas beyond the geographical and environmental space within which the data have been collected. The west coast rivers of New Zealand have a fairly unique character – short, steep rivers, high rainfall, glacial derived flows, relatively intact catchments (at least compared to a lot of New Zealand). I think that compared to places like the Waituna (see Greer et al 2012) or central Waikato (see Franklin et al 2015) strongholds for giant kokopu, or Coromandel (see West et al 2005) or Auckland (see McEwan et al 2009) strongholds for banded kokopu, there are substantial differences in the stream type, flow regimes, and habitat structure where these species are resident. Similarly, there are relatively few places elsewhere in New Zealand where all three kokopu species co-occur in the same way (see Yungnickel 2017). These models may represent the realised niche in this region, but given the different habitat structures and relative differences in species composition and dominance in other regions, I don’t think you provide sufficient evidence to be able to claim that the realised niche elsewhere would be equivalent and thus that your models are transferable. I would like to see some evidence to support this claim – for example how do the environmental gradients you have sampled compare to other regions? While it does not cover extrapolation of mixed effects models, the paper by Booker & Whitehead (2018) may be of interest in this respect.

Fourthly, your site random effect accounts for around half of the variation in biomass in your models of both banded and giant kokopu. Given the general spatial structuring of kokopu species with distance inland and elevation, this is not surprising. However, it again raises the question of transferability of your models to other sites. Why not reduce some of this random error tied to site by explicitly accounting for distance inland/elevation (I appreciate there’s some correlation there!) and potentially other landscape scale variables as fixed effects in a way that will make the results more transferable? Similar approaches using landscape scale predictors have been used to predict fish probability of occurrence across the New Zealand river network previously (e.g. Leathwick et al 2005), so it seems logical that a similar approach here to cover off the broader biogeographical drivers before accounting for local scale variations in biomass associated with the site level variables should be explored.

It is my view that these issues of model structure and transferability need to be better addressed to strengthen this manuscript and improve the usefulness of the models. A few more specific comments follow:

1. I remain unconvinced that your surveys coinciding with spawning season for these species could not have impacted their distribution/behaviour. The Charteris et al manuscript you cite as a justification that kokopu spawn in situ explicitly states that the giant kokopu from that study site migrated out of the reach and then returned following spawning. Similar observations have been observed for giant kokopu by David & Closs 2002. I appreciate that the Franklin et al 2015 paper showed giant kokopu spawned in situ, but this reflects the current level of uncertainty regarding spawning behaviour in spawning kokopu – it’s rarely been documented comprehensively but there’s enough evidence to suggest both aggregation of fish within areas and movement between reaches for spawning.

2. Some previous studies of giant kokopu have demonstrated clear seasonal variation in activity and microhabitat use (e.g. David & Closs 2003). Please address this in your discussion – what does the timing of your surveys mean for the applicability of these models?

3. In all of your tables I think you mistakenly use BK instead of BC to indicate your bank cover fixed effect. If this is not a mistake, I suggest using a different abbreviation for bank cover given that BK is also used to mean banded kokopu in the paper.

4. In your discussion I think it would be helpful to consider what additional value your models have over existing models of fish distribution available for New Zealand (e.g. Leathwick et al 2005). Please compare and contrast the relative strengths and weaknesses of these approaches and, therefore, the suitable contexts for their application.

I remain convinced that there is real promise in this manuscript and with some further work it will merit the hard work that has gone in to collecting the data.

References:

Booker, D. J. and A. L. Whitehead (2018). "Inside or Outside: Quantifying Extrapolation Across River Networks." Water Resources Research 54(9): 6983-7003.

David, B. O. and G. P. Closs (2002). "Behavior of a Stream-Dwelling Fish before, during, and after High-Discharge Events." Transactions of the American Fisheries Society 131(4): 762-771.

David, B. O. and G. P. Closs (2003). "Seasonal variation in diel activity and microhabitat use of an endemic New Zealand stream-dwelling galaxiid fish." Freshwater Biology 48(10): 1765-1781.

Franklin, P. A., et al. (2015). "First observations on the timing and location of giant kōkopu (Galaxias argenteus) spawning." New Zealand Journal of Marine and Freshwater Research 49(3): 419-426.

Greer, M. J. C., et al. (2012). "Complete versus partial macrophyte removal: the impacts of two drain management strategies on freshwater fish in lowland New Zealand streams." Ecology of Freshwater Fish 21(4): 510-520.

Leathwick, J. R., et al. (2005). "Using multivariate adaptive regression splines to predict the distributions of New Zealand's freshwater diadromous fish." Freshwater Biology 50(12): 2034-2052.

McEwan, A. J. and M. K. Joy (2009). "Differences in the distributions of freshwater fishes and decapod crustaceans in urban and forested streams in Auckland, New Zealand." New Zealand Journal of Marine and Freshwater Research 43(5): 1115-1120.

West, D. W., et al. (2005). "Growth, diet, movement, and abundance of adult banded kokopu (Galaxias fasciatus) in five Coromandel, New Zealand streams." New Zealand Journal of Marine and Freshwater Research 39(4): 915-929.

Yungnickel, M.R. (2017) New Zealand’s whitebait fishery: spatial and temporal variation in species composition and morphology. MSc Thesis, University of Canterbury.

7. PLOS authors have the option to publish the peer review history of their article (what does this mean?). If published, this will include your full peer review and any attached files.

Reviewer #2: No

---

## [Author Response · Author response to Decision Letter 1]

21 Jul 2022

Reviewer: Firstly, in my view it is not appropriate to include the biomass of other species as random effect – this should be treated as a fixed effect because, as you state, it is included to account for the direct effect of competitive exclusion.

Author: Interspecific interactions are likely influential, but we have decided to leave the biomass of other kōkopu species as random variables because our aim here was to examine kōkopu associations with abiotic habitat variables, so we only need to account for biotic influences. Investigating biotic interactions between kōkopu was definitely not an aim and isn’t necessary to predict local kōkopu abundance from habitat features. Furthermore, having the biomass of the other kōkopu species as random variables allows our partial plots to provide the most accurate estimation of each species’ habitat use by accounting for the biomass of other kōkopu present (L227-230). An additional benefit of focusing on abiotic variables is that the models can be used to estimate expected kōkopu biomass from local habitats without spotlighting them, which would otherwise be needed if other species were added as fixed variables in the model (L477-480).

Reviewer: Secondly, it appears that you are treating all sampled reaches as independent sites, even though they are nested within catchments and hence subject to correlated effects. In my view this is potentially an inappropriate treatment of pseudo-replication and the spatial structuring of reaches by catchment needs to be accounted for in your formulation of your site random effect (e.g. reaches need to be nested within catchments).

Author: L220-227: Reaches were used as replicates (for more statistical power estimating habitat-biomass relationships) and a random stream factor (3 reaches within a stream) was added to account for within stream variability. This method provides the strongest statistical inference and accounts for the adjacency of reaches. Including a random catchment factor is unnecessary because there were few cases where streams were found in the same catchment, and would lead to further analytical problems due to uneven replication among catchments. Finally, we have already accounted for key factors that would likely cause catchment variation by ensuring there were no downstream barriers, all reaches and streams were accessible to each species of kōkopu, and selecting catchments that were minimally impacted.

Reviewer: Thirdly, I am not yet sufficiently convinced of the transferability of your models to areas beyond the geographical and environmental space within which the data have been collected. The west coast rivers of New Zealand have a fairly unique character – short, steep rivers, high rainfall, glacial derived flows, relatively intact catchments (at least compared to a lot of New Zealand). I think that compared to places like the Waituna (see Greer et al 2012) or central Waikato (see Franklin et al 2015) strongholds for giant kokopu, or Coromandel (see West et al 2005) or Auckland (see McEwan et al 2009) strongholds for banded kokopu, there are substantial differences in the stream type, flow regimes, and habitat structure where these species are resident. Similarly, there are relatively few places elsewhere in New Zealand where all three kokopu species co-occur in the same way (see Yungnickel 2017). These models may represent the realised niche in this region, but given the different habitat structures and relative differences in species composition and dominance in other regions, I don’t think you provide sufficient evidence to be able to claim that the realised niche elsewhere would be equivalent and thus that your models are transferable. I would like to see some evidence to support this claim – for example how do the environmental gradients you have sampled compare to other regions? While it does not cover extrapolation of mixed effects models, the paper by Booker & Whitehead (2018) may be of interest in this respect.

Author: Our models estimate biomass from minimally impacted catchments with no migratory impediments. That is entirely appropriate since the goal of our study was to evaluate ‘local’ influences and other landscape scale influences on distribution would confound our patterns. We have strengthened the Introduction, Methods and Discussion text to further clarify that point and acknowledge the need to account for other landscape-scale drivers (e.g., L97-99, L153; L472-474; L480-484). Furthermore, the Conclusion emphasises that these estimations will thus provide a baseline that should indicate if kōkopu biomass is influenced by other environmental influences (i.e., a barrier; L494-496). These estimations should not be biased by our site selection because they were not glacially derived, abnormally steep or elevated (>100 m elevation), and spanned up to 17km inland (to ensure we did not exclude giant kōkopu; L151-153). Nevertheless, we now acknowledge in the manuscript that long-term widespread monitoring is required to confirm our patterns (488-490). Although the West Coast does have high rainfall, we do not believe this would notably influence our local habitat results because tagged kōkopu in these streams take refuge during the floods and typically remain within the same reaches (unpublished data). Finally, constructing a model like the one being asked for requires repeating a study like this in different areas so the large-scale influences were properly incorporated. Such an approach would also need to encompass the range of landscape-scale influences (e.g., by incorporating habitats close to and far from the sea). We clearly do not have the data to do. Moreover, given realistic resourcing constraints, we have actually done the best thing to produce the most applicable local model by working in least impacted catchments and thereby controlling for most other influences that would confound local habitat use.

Author: Regarding species overlap, Yungnickel’s (2017) Figure 2.1 (assuming that is the figure being referenced) shows considerable kōkopu distribution overlap, particularly across the north and west coast of both main islands. However, as long as the models are applied within the targeted kōkopu species’ natural range, they should provide an accurate standardised estimation (regardless of which other kōkopu species are present) due to incorporating reaches that included a variety of kōkopu co-occurrence combinations. We now mention this in the text and provide a Table in supplementary materials that describes the range of different assemblages our data encompass (S1 Table; L474-484). Moreover, including the density of other kōkopu as a random effect in the models parses this variance, thereby providing generally applicable local models.

Reviewer: Fourthly, your site random effect accounts for around half of the variation in biomass in your models of both banded and giant kokopu. Given the general spatial structuring of kokopu species with distance inland and elevation, this is not surprising. However, it again raises the question of transferability of your models to other sites. Why not reduce some of this random error tied to site by explicitly accounting for distance inland/elevation (I appreciate there’s some correlation there!) and potentially other landscape scale variables as fixed effects in a way that will make the results more transferable? Similar approaches using landscape scale predictors have been used to predict fish probability of occurrence across the New Zealand river network previously (e.g. Leathwick et al 2005), so it seems logical that a similar approach here to cover off the broader biogeographical drivers before accounting for local scale variations in biomass associated with the site level variables should be explored.

Author: This argument is already mostly covered above. We agree that large scale factors are important to consider and certainly account for large amounts of variation in some places, but we focus on local habitat variables in least impacted catchments to provide the best possible model of local habitat conditions. Our sites didn’t cover great distances inland (max 17 km), elevations (<100m), and were not abnormally steep, so those influences have been controlled for. As already described, we now highlight that our study is focused on evaluating kokopu abundance at a local scale and is not about predicting the spatial distribution of kokopu across the landscape. Thus, further confusion about the aims should be avoided. Moreover, in the absence of a mega kōkopu study involving collection of local habitat use across many regions, our models provide the best and most applicable evaluations of local habitat use because we have controlled for the landscape-scale influences the reviewer is concerned about. For example, if a local manager anywhere in the country wants to construct the best local kōkopu habitat, they can use our results (the landscape features will control whether kōkopu get there, but they’ll still have appropriate habitat). Finally, we’re not going to change the paper’s aim away from local influences because our data do not provide any strength of inference to assess those (i.e., they were collected in least impacted catchments close to the coast and at low altitude, so there in no ability to assess those effects).

Reviewer: It is my view that these issues of model structure and transferability need to be better addressed to strengthen this manuscript and improve the usefulness of the models. A few more specific comments follow:

Reviewer: 1. I remain unconvinced that your surveys coinciding with spawning season for these species could not have impacted their distribution/behaviour. The Charteris et al manuscript you cite as a justification that kokopu spawn in situ explicitly states that the giant kokopu from that study site migrated out of the reach and then returned following spawning. Similar observations have been observed for giant kokopu by David & Closs 2002. I appreciate that the Franklin et al 2015 paper showed giant kokopu spawned in situ, but this reflects the current level of uncertainty regarding spawning behaviour in spawning kokopu – it’s rarely been documented comprehensively but there’s enough evidence to suggest both aggregation of fish within areas and movement between reaches for spawning.

Author: L144-147: We know of only one source suggesting that ripe giant kōkopu migrate downstream in McDowall (1990); there was no further information or source for these observations, so they are speculative. McDowall (1990) also states that ripe and spent banded and giant kōkopu are found in typical adult habitats and will likely not migrate far to spawn. Additionally, Charteris (2002) found that shortjaw and other galaxiid species generally stayed within their resident home ranges over the spawning season. When fish did move out of their home ranges, it was not thought that it was a direct result of spawning, as it was not limited to spawning time. Thus, we don’t think the movement argument stacks up. Furthermore, we did not find fish in a condition that suggested they had recently spawned or were about to spawn.

Charteris, S. C. (2002). Spawning, egg development and recruitment of diadromous galaxiids in Taranaki, New Zealand. MSc. thesis, Massey University, Palmerston North, New Zealand.

McDowall, R. M. (1990). New Zealand freshwater fishes: a natural history and guide: Auckland, Heinemann Reed.

Reviewer: 2. Some previous studies of giant kokopu have demonstrated clear seasonal variation in activity and microhabitat use (e.g. David & Closs 2003). Please address this in your discussion – what does the timing of your surveys mean for the applicability of these models?

Author: L484-488: We now acknowledge that patterns may vary seasonally due to small kōkopu immigration events in spring, and the subsequent gain and loss of large kokopu biomass associated with spawning in autumn. However, it is unlikely that seasonal shifts in kōkopu microhabitat use (i.e., position within a pool) will influence our local habitat associations substantially because fish generally remain within the same relatively short home range (David & Closs, 2003).

Reviewer: 3. In all of your tables I think you mistakenly use BK instead of BC to indicate your bank cover fixed effect. If this is not a mistake, I suggest using a different abbreviation for bank cover given that BK is also used to mean banded kokopu in the paper.

Author: This was a mistake and has now been fixed.

Reviewer: 4. In your discussion I think it would be helpful to consider what additional value your models have over existing models of fish distribution available for New Zealand (e.g. Leathwick et al 2005). Please compare and contrast the relative strengths and weaknesses of these approaches and, therefore, the suitable contexts for their application.

Author: L469-480: The advantages of our localised models over pre-existing New Zealand fish distribution models is that they (1) provide standardised biomass estimations for small and large size-classes instead of species presence; (2) were developed using consistent methodology at an explicitly local spatial scale; and (3) avoided confounding influences by only included streams with no downstream barriers, invasive species, or notable degradation. Furthermore, these standardised predictions are applicable to kōkopu populations across New Zealand, regardless of which other kōkopu species are present, because they account for the influence of other kōkopu species and were developed using reaches that included most combinations of kōkopu co-occurrence (S1 Table). By accounting for biotic interactions between kōkopu species in model development, more accurate standardised biomass estimations can be made using local habitat features without needing to spotlight stream reaches. 

L483-484: We have noted the limitations of our models, in that they (1) can only be applied to reaches that are within the natural distribution of the target kōkopu species, and (2) may vary regionally, particularly when applied to areas less accessible to kōkopu or with more impacted catchments. However, as stated above and throughout the manuscript, the models will work most effectively when other factors are accounted for. Furthermore, they are ideal for identifying the best local habitat for managers to construct anywhere (as described above).

---

## [Decision Letter · Decision Letter 2]

5 Oct 2022

PONE-D-21-39491R2Predicting biomass of resident kōkopu (*Galaxias*) populations using local habitat characteristicsPLOS ONE

Dear Dr. Crichton,

Thank you for submitting your manuscript to PLOS ONE. After careful consideration, we feel that it has merit but does not fully meet PLOS ONE’s publication criteria as it currently stands. Therefore, we invite you to submit a revised version of the manuscript that addresses the points raised during the review process.

It appears most of the comments concerns raised in previous submissions have been addressed. However, the authors have only made critical counterpoints to some (in their response to reviewers) and these editorial changes remain to be fully addressed or rebutted.  These are minor points and should be able to be easily addressed by the authors without much trouble. Please refer to your "response to reviewers" for editorial remarks that you have chosen not to change.

We look forward to receiving your revised manuscript.

Kind regards,

Madison Powell, PhD

Academic Editor

PLOS ONE

Journal Requirements:

Additional Editor Comments:

It appears most of the comments concerns raised in previous submissions have been addressed. However, the authors have only made critical counterpoints to some (in their response to reviewers) and these editoral changes remain to be addressed.

Reviewers' comments:

Reviewer's Responses to Questions

**Comments to the Author**

1. If the authors have adequately addressed your comments raised in a previous round of review and you feel that this manuscript is now acceptable for publication, you may indicate that here to bypass the “Comments to the Author” section, enter your conflict of interest statement in the “Confidential to Editor” section, and submit your "Accept" recommendation.

Reviewer #2: (No Response)

2. Is the manuscript technically sound, and do the data support the conclusions?

Reviewer #2: Partly

3. Has the statistical analysis been performed appropriately and rigorously? 

Reviewer #2: No

4. Have the authors made all data underlying the findings in their manuscript fully available?

Reviewer #2: Yes

5. Is the manuscript presented in an intelligible fashion and written in standard English?

Reviewer #2: Yes

6. Review Comments to the Author

Reviewer #2: Dear Authors,

It is great to see the ongoing improvements to this manuscript in response to prior review comments. It retains its strong empirical foundation and in acknowledging and addressing some of the limitations around transferability and application relative to other species distribution modelling approaches I think now gives a more balanced view of its applied value. However, I believe you have not appropriately addressed the errors in the structure of your statistical models as highlighted by previous reviews.

The structure of the model (in terms of what is included as fixed v random effects) should not be determined by convenience to support a desired narrative. It must be dictated by the protocols of the statistical methodology. In that respect, you must include the biomass of other species as a fixed effect. It is a continuous variable, the value of which has real meaning, and as such can only be treated as a fixed effect. My understanding is that random effects must be factors and that the levels of those factors should effectively be ‘meaningless’ in so far that you can use any set of labels and it doesn’t change the interpretation of what it means (e.g. labelling sites 1, 2, 3 versus A, B, C). As such, I’m not even sure how you’ve included biomass as a random variable as it presumably would have had to be converted to a factor somehow (maybe high/medium/low?), but even if you’ve done this the factor labels have meaning – they’re ordered – and so do not meet the normal rules of defining a random effect. I appreciate that this may appear ‘inconvenient’ for how you are wanting to frame your narrative, but assuming that biomass is retained as a significant fixed effect in your models, then I think it can potentially help with the transferability of the models – it would allow someone who is trying to apply the model in the real world to set biomass of a species at zero, if for example, it was known not to co-exist at the site. Similarly, you could provide median and/or confidence intervals around your observed biomass values that someone wishing to use the model could utilise to get an understanding of the range of biomass in the target species given the local habitat conditions. My key point is that just by including biomass of competing species as a fixed effect, it doesn’t negate the value of the model – in fact I would argue it strengthens the potential utility because it allows users to more appropriately recognise uncertainty.

Another component of your GLMM that you need to address is the treatment of site/reach. Your use of the term ‘site’ as the term in the model appears to have created some confusion; to most people at first glance this would be interpreted as the lowest level of replication, but I think in this case this is what you term a ‘reach’. The prior review comments remain valid in this respect in so far that it would be most appropriate for ‘reach’ to be nested within ‘site’ within your model formulation. This will appropriately address the spatial structuring of your data and currently inappropriate treatment of pseudo-replication.

More generally, I feel that the comments made in the previous review around the uncertainty of the potential impacts of spawning remain valid – I appreciate your responses, but you rightly acknowledge that there remains considerable uncertainty over the role of migratory movements around spawning. As such, I feel that you could provide a more balanced acknowledgement of this in your manuscript. It’s not a fundamental flaw, but there appears to be enough evidence to suggest that particularly for giant kokopu, at least in some streams, they likely undertake spawning migrations and at the moment you largely dismiss this possibility – just be honest about the uncertainty.

I feel that there is a strong manuscript here, but please don’t ignore good statistical practice for the sake of fulfilling a particular narrative. Likewise, it is always advisable to provide a balanced and honest discussion of the limitations/uncertainty in your work – this improves its credibility and is not something to be afraid of.

Best of luck with completing your manuscript.

7. PLOS authors have the option to publish the peer review history of their article (what does this mean?). If published, this will include your full peer review and any attached files.

Reviewer #2: No

---

## [Author Response · Author response to Decision Letter 2]

11 Dec 2022

Reviewer #2: Dear Authors,

Reviewer #2: It is great to see the ongoing improvements to this manuscript in response to prior review comments. It retains its strong empirical foundation and in acknowledging and addressing some of the limitations around transferability and application relative to other species distribution modelling approaches I think now gives a more balanced view of its applied value. However, I believe you have not appropriately addressed the errors in the structure of your statistical models as highlighted by previous reviews.

Reviewer #2: The structure of the model (in terms of what is included as fixed v random effects) should not be determined by convenience to support a desired narrative. It must be dictated by the protocols of the statistical methodology. In that respect, you must include the biomass of other species as a fixed effect. It is a continuous variable, the value of which has real meaning, and as such can only be treated as a fixed effect. My understanding is that random effects must be factors and that the levels of those factors should effectively be ‘meaningless’ in so far that you can use any set of labels and it doesn’t change the interpretation of what it means (e.g. labelling sites 1, 2, 3 versus A, B, C). As such, I’m not even sure how you’ve included biomass as a random variable as it presumably would have had to be converted to a factor somehow (maybe high/medium/low?), but even if you’ve done this the factor labels have meaning – they’re ordered – and so do not meet the normal rules of defining a random effect. I appreciate that this may appear ‘inconvenient’ for how you are wanting to frame your narrative, but assuming that biomass is retained as a significant fixed effect in your models, then I think it can potentially help with the transferability of the models – it would allow someone who is trying to apply the model in the real world to set biomass of a species at zero, if for example, it was known not to co-exist at the site. Similarly, you could provide median and/or confidence intervals around your observed biomass values that someone wishing to use the model could utilise to get an understanding of the range of biomass in the target species given the local habitat conditions. My key point is that just by including biomass of competing species as a fixed effect, it doesn’t negate the value of the model – in fact I would argue it strengthens the potential utility because it allows users to more appropriately recognise uncertainty.

Authors: L224-230: We originally included the biomass of other kōkopu size classes as fixed effects to test for potentially confounding interspecific interactions, but these were removed because we found no associations between kōkopu of the same size class (S1 Fig, S1 Table). Furthermore, any direct effect of kōkopu biomass on the response, particularly between size classes, would likely be indirectly driven by abiotic variables, and we are unable to effectively disentangle abiotic and biotic effects. To effectively isolate biotic and abiotic effects, we would need to include interactions between each kōkopu species’ size class and each habitat variable, which would require us to not only cover a wide array of different abiotic conditions, but also replicate this abiotic array with different kōkopu co-occurrences, which would require a naturally unrealistic dataset that we do not possess. Moreover, including the biomass of other size classes as an additive effect would lead to misleading habitat associations because of the confounding effects between abiotic and biotic predictors (as stated above). Therefore, we have not included biotic variables in our models at all.

*Added because this is relevant to early review comments about kōkopu grouping*

Authors: L208-212: We have combined the biomass of the small kōkopu size classes because juvenile shortjaw and giant kōkopu were absent from most reaches due to being naturally rare, which meant we could not develop effective habitat-biomass models for these size classes. However, this grouping should not confound our results because juvenile kōkopu likely occupy the same habitats due to being competitively displaced by larger dominant congeners. We found that our partial plots and models became slightly more accurate because of this grouping, indicating that the small number of juvenile giant and shortjaw kōkopu present were likely using similar habitats to small banded kōkopu.

Reviewer #2: Another component of your GLMM that you need to address is the treatment of site/reach. Your use of the term ‘site’ as the term in the model appears to have created some confusion; to most people at first glance this would be interpreted as the lowest level of replication, but I think in this case this is what you term a ‘reach’. The prior review comments remain valid in this respect in so far that it would be most appropriate for ‘reach’ to be nested within ‘site’ within your model formulation. This will appropriately address the spatial structuring of your data and currently inappropriate treatment of pseudo-replication.

Authors: We believe the manuscript adequately explains our spatial structuring and used terms. We firstly introduce our spatial structuring in the beginning of our methods: 

“To investigate which habitat features are most strongly associated with reach-scale kōkopu biomass, three 50-m reaches were sampled within each of 19 streams…” (L144-145). 

Then in our analytical methods, we introduce and explain the ‘site’ term and our site/reach spatial structuring in an analytical context: 

“Linear mixed-effects models were constructed using the ‘lmer’ function and included a random factor for stream site, hereafter referred to as ‘site’. A random factor for site was included so that each of the three reaches nested within each site could be used independently to examine how habitat characteristics influenced kōkopu biomass.” (L231-236).

Authors: Because each reach was not sampled more than once within each stream, and the three reaches sampled per stream are the only pseudo-replicates we have for each model, only a random variable for site is needed. A random variable for reach nested within site does not change our results, because there is no variability within each reach (one sample per reach). A random effect for reach could only be included if we used one global model that tested for habitat associations between classes and species, meaning each measurement of biomass for each species and class would be a pseudo-replicate. However, because we have independent models for each size class of each species, we cannot do this. We created this global model to test if a random reach effect was influential, but it explained 0% of our random/error variation. We have chosen to stick with independent models because 1) we can obtain an R2 for each size class, 2) the partial plots for independent models are far more informative, and 3) because we can test for relative variable importance for each case.

Reviewer #2: More generally, I feel that the comments made in the previous review around the uncertainty of the potential impacts of spawning remain valid – I appreciate your responses, but you rightly acknowledge that there remains considerable uncertainty over the role of migratory movements around spawning. As such, I feel that you could provide a more balanced acknowledgement of this in your manuscript. It’s not a fundamental flaw, but there appears to be enough evidence to suggest that particularly for giant kokopu, at least in some streams, they likely undertake spawning migrations and at the moment you largely dismiss this possibility – just be honest about the uncertainty.

Authors: L146-151: We have now emphasised in the manuscript that there is uncertainty around kōkopu spawning behaviours and giant kōkopu spawning migrations. But as previously stated, we cannot find any evidence to suggest kōkopu migrate for spawning, whereas our cited studies state that all three species of kōkopu generally remained within their home range or did not move far to spawn. If you have a link to any of this evidence we will readily include it.

Reviewer #2: I feel that there is a strong manuscript here, but please don’t ignore good statistical practice for the sake of fulfilling a particular narrative. Likewise, it is always advisable to provide a balanced and honest discussion of the limitations/uncertainty in your work – this improves its credibility and is not something to be afraid of.

Best of luck with completing your manuscript.

---

## [Decision Letter · Decision Letter 3]

20 Feb 2023

Predicting biomass of resident kōkopu (*Galaxias*) populations using local habitat characteristics

PONE-D-21-39491R3

Dear Dr. Crichton,

 We’re pleased to inform you that your manuscript has been judged scientifically suitable for publication and will be formally accepted for publication once it meets all outstanding technical requirements.

Kind regards,

Ram Kumar, Ph.D.

Academic Editor

PLOS ONE

---

## [Editor Report · Acceptance letter]

6 Mar 2023

PONE-D-21-39491R3 

Predicting biomass of resident kōkopu (*Galaxias*) populations using local habitat characteristics 

Dear Dr. Crichton:

I'm pleased to inform you that your manuscript has been deemed suitable for publication in PLOS ONE. Congratulations! Your manuscript is now with our production department. 

Kind regards, 

on behalf of

Professor Ram Kumar 

Academic Editor

PLOS ONE